# Bio-Fertilizers Based on Digestate and Biomass Ash as an Alternative to Commercial Fertilizers—The Case of Tomato

**Katarzyna Przygocka-Cyna** [1], **Przemysław Barłóg** [1], **Tomasz Spiżewski** [2,*] **and Witold Grzebisz** [1]

1    Department of Agricultural Chemistry and Environmental Biogeochemistry, Poznań University of Life Sciences, Wojska Polskiego 38/42, 60-625 Poznań, Poland; katarzyna.przygocka-cyna@up.poznan.pl (K.P.-C.); przemyslaw.barlog@up.poznan.pl (P.B.); witold.grzebisz@up.poznan.pl (W.G.)

2    Department of Vegetables Production, Poznan University of Life Sciences, Wojska Polskiego 71F, 60-625 Poznań, Poland

*    Correspondence: tomasz.spizewski@up.poznan.pl

**Abstract:** The reutilization of agricultural wastes, as bio-fertilizers, is the key way to close the nutrient cycle and save mineral fertilizers. This hypothesis was verified in three consecutive seasons, treating tomato with three bio-fertilizers on the background of a standard rate of mineral fertilizer. The bio-fertilizers differed in their C:N ratio, which was 13:1, 21:1, and 6:1 for the A, B, and C fertilizers, respectively. They were applied at the rate of 200, 400, 800, and 1600 kg ha$^{-1}$. The average fruit yield increased in the order: B < C < A. For the relevant fertilizer, the maximum commercial yield was 91, 87, and 101 t ha$^{-1}$, for a respective rate of 1600, 200, and 400 kg ha$^{-1}$. The number of fruits (CFN), as the dominant yield component, indirectly reflected the nitrogen (N) supply to plants. A shortage or excess of N on plots treated with the B or C fertilizers, resulted in a decreased CFN, leading to a yield decline. The year-to-year variability in the potassium (K) content reflected fairly well the variable weather conditions, responsible for water management by tomato. The conducted study showed that the tested bio-fertilizers can replace mineral fertilizer, as long as they are applied at well-defined rates.

**Keywords:** biomass ash; digestate; fruit number per unit area; heavy metals; nutrients; phytochemicals; total fruit yield

## 1. Introduction

The European Bio-economy Strategy 2018–2030 is based on five points: (i) ensuring food and nutrition security, (ii), managing natural resources in a sustainable way (iii) reducing dependence on non-renewable, unsustainable resources, whether sourced domestically or from abroad, (iv) mitigating and adapting to climate change, (v) maintaining European competitiveness and creating new jobs [1]. In 2019, energy produced from renewable sources shared 19.7% of total energy consumed in the EU-27, i.e., 0.3% below the target of 20% [2]. In Poland, the contribution of energy from renewable sources in total energy consumption reached 12.2% in 2019, i.e., below the target of 15% [2,3]. Revised Renewable-Energy Directive-RED II, (2018/2001EU) assumes, in accordance to points no. 2 and 3, that each EU state member will produce 32% of gross energy from renewable sources by 2030 [4].

Biomass is considered in the European Union (EU), as presented in RED II, as the key source of renewable energy; it would allow carbon neutrality to be achieved by 2050. Its widespread use in the European Union (UE) would substantially decrease the requirement for non-renewable energy carriers, such as coal and oil [5]. At present, the main sources of biomass used for energy production in Europe are solid biomass and its residues, which delivers about 80% of produced energy in total [6]. The market potential for biomass production for energy in Poland is 15.5 million t annually, of which $^1/_3$ can be covered by straw (rap, cereals) [5,7]. This amount of combusted biomass, based on various sources,

results in 0.4–0.6 million t year$^{-1}$ of ash. This is a substantial amount of residue, which needs to be managed in an efficient way. The best solution, taking into account RED II, is to use biomass ash as a lime or a source for mineral fertilizer production.

The secondary challenge, as important as the primary one, is to develop an efficient strategy for ash management [8]. The characteristics of ash should be considered from four perspectives, at least. The first is its mass left after biomass combustion. This ash feature is plant species dependent, varying from 1 to more than 10%. The second is its chemical composition, dominated, in general, by alkaline cations. The order of this element's content in ashes from wood is as follows: Ca > K > Mg > S > P. This property of biomass ashes allows their direct use as fertilizers or as a source for production of mineral fertilizers. The content of calcium and other basic cations is a factor that has an impact on the pH of ashes, which ranges from 10 to 12. This high pH is the prerequisite for the use of biomass ashes as a soil amendment, replacing lime. Biomass ash, depending on plant species, is also rich in other nutrients, including trace elements. Its end-use advantage as fertilizer is limited by the content of heavy metals, mainly lead, cadmium and arsenic [9,10].

Another method of biomass utilization for energy production is its transformation, under strict anaerobic conditions, into a mixture of two gases, $CH_4$ and $CO_2$, termed as biogas. Europe is a leader in biogas production, but its share in energy production from renewable sources is still low [11,12]. The liquid residue of anaerobic biomass digestion is biogas digestate, frequently called digestate (D) [13]. Digestate is rich, in spite of its high variability in concentration, in all nutrients, required for plant growth, except sulfur. Therefore, raw digestate slurry is most frequently directly used as a liquid fertilizer [14]. Anaerobic digestion (AD) is therefore, not only a technological solution, but also an economic instrument towards an active incorporation of agriculture into a circular bioeconomy [15]. The challenge for policy makers, as in the case of biomass ash, is to develop a sound strategy for the effective disposal of digestate. Among numerous solutions, the easiest one seems to be to treat digestate as a slurry and spread it directly on fields [16]. However, this approach to digestate management is costly and non-allowed in winter [17,18]. An alternative solution is to use dewatered digestate as a source for the production of organic fertilizers (bio-fertilizers) [19–21]. The disadvantages of solids separation from the raw slurry are both the high-energy needs and a loss of ammonia [14].

A use of wastes, such as biogas digestate and biomass ash in agriculture fulfils the objectives of the circular bioeconomy concept [15]. The current study clearly showed that bio-fertilizers can exchange or decrease the use of mineral fertilizers to a great extent [22]. The key element of bio-fertilizer action depends on their impact on soil organic matter mineralization and the availability of plant nutrients. Application of organic or organomineral fertilizers based on digestate, characterized by a narrow C:N ratio, can result in soil depletion with macronutrients such as potassium. magnesium and even copper [20]. Vegetables are an important source of vitamins, minerals, and phytochemicals, positively affecting human health [23]. Due to its nutritional and health benefits tomato is one of the world's most widely cultivated vegetables [24]. Tomato production in Europe was 18 mln t in 2018, contributing to 10% of the world production (181 mln t).

Poland is one of the top vegetable producers in Europe. Vegetables are frequently treated as test crops for the evaluation of bio-fertilizers, originating from recycled materials, such as ashes or digestate [25,26]. Lead and cadmium are toxic for humans when present in excessive amounts in edible parts of crop plants [27,28]. The maximum acceptable levels of contents of harmful metals in food, including edible parts of vegetables, are defined as standards, which should be used to evaluate the use of any wastes or bio-fertilizers as soil amendments [29]. The advantage of tomato over other vegetables is that its fruits do not accumulate heavy metals. The main reason for this is a very low level of transmission factor, frequently below 0.1 [30,31].

The assessment of the fertilizing value of bio-fertilizers produced from agricultural waste is not simple and unambiguous. This is a classic example of *a black box*. The chemical composition of this type of fertilizer is not a direct criterion for assessing their impact

on plant growth, yield and its nutritional value [32,33]. The response of the crops is due to both straight and indirect action [34,35]. Bio-fertilizer is a source of nutrients for the plant, but their content in the directly available form is usually low [14]. The indirect effects of bio-fertilizers are mainly due to their impact on soil processes [13]. For these reasons, the validation of this group of fertilizers requires taking into account three groups of criteria: (a) production, (b) ecological, (c) human health [20,27,33]. The production criteria the validation of this group of fertilizers include: (a) yield, (b) chemical composition of edible plant parts, including (i) content of organic compounds, (ii) content of nitrogen and other nutrients, and (iii) content of heavy metals [20,36]. Ecological criteria concern the influence of bio-fertilizer on (a) the rate soil organic matter mineralization, (b) the content and availability of main nutrients, (c) the content and availability of heavy metals, (d) heavy metals transfer from soil to plant [37,38]. The risk assessment of bio-fertilizer impact on human health result both from the content of undesirable components, mainly heavy metals in edible parts of plants, and the amount of their consumption [39]. The heavy metal content of the edible parts of vegetables undergoes a rigorous control [40].

It was hypothesized that (1) the biomass ash and digestate bio-fertilizer products are suitable in application to field grown tomato, (2) the bio-fertilizers can be used as the basic fertilizer source for tomato plants, replacing a standard mineral fertilizer, (3) the bio-fertilizer cauls improve the quality of tomato fruits, (4) the bio-fertilizer cannot create a threat for human health.

The objective of the study was to evaluate the fertilizer value of three bio-fertilizers substantially different in proportion in the used components, i.e., biomass ash and digestate and some other additives, affecting the C:N ratio. The effect of the tested bio-fertilizers on tomato yield, yield components, contents of organic compounds and minerals in fruits was evaluated on the background of the effect of a standard mineral fertilizer.

## 2. Materials and Methods

### 2.1. Site Description

Field tests on tomato (*Solanum lycopersicum* L.) response to bio-fertilizers based on biomass ash and digestate from maize silage were conducted in three consecutive years, i.e., 2016, 2017, and 2018, at the Experimental Farm Marcelin of the Poznan University of Life Sciences, Poland (52°24′ N, 16°51′ E). The field experiment was established on soil originating from loamy sand, lying on sandy loam, and classified as Albic Luvisol. Soil pH was variable, ranging from acid in 2016 to slightly acid in the other two years. The content of mineral nitrogen ($N_{min}$) showed low variability, and $NO_3$-N was a dominant N form. Soil fertility level in all years was, in general, high. The content of available phosphorus (P), was very high. Contents of K and calcium (Ca) were in the good class, and magnesium (Mg) in the high class (Table 1). The content of micronutrients and selected heavy metals is presented in Table 2.

**Table 1.** Topsoil agrochemical properties before tomato planting-macronutrients.

| Year | pH [1] | P [2] | K [2] | Mg [2] | Ca [2] | $NO_3$-N [4] | $NH_4$-N [4] | $N_{min}$ [5] |
|------|--------|-------|-------|--------|--------|--------|--------|--------|
| | | | | mg kg$^{-1}$ | | | kg ha$^{-1}$ | |
| 2016 | 5.4 | 440 [VH3] | 230 [G] | 200 [H] | 2800 [G] | 28 | 6 | 34 |
| 2017 | 5.5 | 409 [VH] | 280 [G] | 222 [H] | 2650 [G] | 24 | 14 | 38 |
| 2018 | 5.6 | 390 [VH] | 260 [G] | 266 [H] | 2550 [G] | 24 | 10 | 34 |

[1] 1 M KCl; [2] Mehlich 3 [40]; [3] ranges [41]: [VH]—very high; [H]—high; [G]—good; [4] 0.01 M CaCl$_2$; [5] $N_{min}$ = $NO_3$-N + $NH_4$-N.

### 2.2. Weather Conditions

The local climate, classified as intermediate between Atlantic and Continental, is seasonally variable. The total sum of precipitation during the tomato growing season (May-September) was 236.6 mm in 2016, 434 mm in 2017, and 234.4 mm in 2018, whereas the long-term average is 285 mm (Table 3). The key environmental disadvantage during tomato

vegetation was low precipitation in August, concomitant with elevated temperatures. In 2018, the hottest year of the study, the amount of precipitation in August reached only 10 mm, but the average temperature was higher by 3 °C compared to the long-term average.

**Table 2.** The content of micronutrients and heavy metals in the topsoil.

| Year | Fe [1] | Mn [1] | Zn [1] | Cu [1] | Pb [1] | Cd [1] |
|------|------|------|------|------|------|------|
|      |      |      | mg kg$^{-1}$ |      |      |      |
| 2016 | 370 [G] | 86 [G] | 44 [H] | 2.8 [G] | 7.1 | 0.12 |
| 2017 | 340 [G] | 92 [G] | 40 [H] | 3.2 [G] | 6.9 | 0.10 |
| 2018 | 330 [G] | 77 [G] | 42 [H] | 3.0 [G] | 6.5 | 0.09 |

[1] Mehlich 3 [32]; ranges [34]: [H]—high; [G]—good.

**Table 3.** Weather conditions during consecutive years of tomato cultivation.

| Years/Months | May | June | July | August | September |
|------|------|------|------|------|------|
| 2016 | 15.71/32.62 | 18.8/71.5 | 19.3/102.9 | 18.0/25.4 | 17.5/3.2 |
| 2017 | 15.2/49.9 | 18.3/86.1 | 18.4/164.2 | 19.6/109.9 | 13.8/23.9 |
| 2018 | 19.0/30,9 | 19.3/49.9 | 20.8/97.2 | 21.5/11.0 | 16.0/45.4 |
| 1961–2018 | 13.7/44 | 17.1/56 | 18.8/74 | 18.5/63 | 13.4/38 |

State synoptic station at Ławica-Poznań.

*2.3. Experimental Design*

A one factorial field experiment was arranged as a randomized complete block design, replicated three-fold, comprising 12 treatments based on three bio-fertilizers applied in four rates and two control treatments:

(1)　Bio-Fertilizer A (composed of 70% of biomass ash (BA) + 25% of digestate (D) + 5% of S$^0$);

(2)　Bio-Fertilizer B (composed of 25% of BA + 75% of digestate D);

(3)　Bio-Fertilizers C (composed of 30% BA + 45% of digestate D) + 5% of S$^0$ + 5% of urea + 15% of phosphoric rock;

(4)　The rate of applied bio-fertilizers was as follows: 200; 400; 800, and 1600 kg ha$^{-1}$;

(5)　Absolute Control plot (AC)—i.e., plot without application of any fertilizers;

(6)　Mineral Control plot (MC)—i.e., plot fertilized with Yara Mila Complex (N-12%, P$_2$O$_5$-11%; K$_2$O-18%; MgO-2.7%; SO$_3$-2%; B-0.015%; Fe-0.2%; Mn-0.02%; Zn-0.02%) supplied at an N rate of 100 kg N ha$^{-1}$.

The chemical composition of the tested bio-fertilizers is shown in Table 4 and the amount of applied N in Table 5.

All fertilizers, including Yara Mila, were applied two weeks before planting of tomato seedlings. The tomato variety *Polbig* was planted between 15 and 25 of May each year. Ten plants were planted on a plot of 4 m$^{-2}$. Plants were managed in accordance with the codex of good agricultural practice. Each year, yellow lupine was a preceding crop for tomato. Tomato fruits were hand harvested during seven consecutive weeks at weekly intervals, from the beginning of August from the whole plot area. The commercial yield (CY) was determined by removing fruits out of the marketable norm (small, wounded, diseased). The following yield factors were analyzed: total weight of one fruit (TFW), commercial weight of one fruit (CFW), total number of fruits per m$^2$ (TFN) and commercial number of fruits per m$^2$ (CFN).

*2.4. Chemical Measurements*
2.4.1. Soil

Composite soil samples (0–30; 30–60 cm) for mineral N (N$_{min}$) determination were collected at the beginning of the growing season. For N$_{min}$ determination 20 g of soil samples were shaken for 1 h with 100 mL of a 0.01-M CaCl$_2$ solution (soil/solution ratio 5:1; m/v). Concentrations of mineral N forms (NH$_4$-N and NO$_3$-N) were determined

by the colorimetric method using flow injection analyses (FIAstar5000, FOSS, Hilleroed, Denmark). For determination of available forms of nutrients (P, K, Mg, Ca, Fe, Mn, Zn, and Cu), available cadmium(Cd), and lead(Pb) were collected at the beginning of the growing season. The soil samples were then air-dried and crushed to pass a sieve of 2-mm mesh size. The extractable nutrients and heavy metals were determined, using the Mehlich 3 method [40–42]. The content of available P in the extract was determined calorimetrically, while the content of K, Mg and Ca, Fe, Mn, Zn, Cu, Pb, Cd and Ni was determined, using a flame type atomic absorption spectrometer FAAS, (Atomic SpectrAA-55B, Sn. Clara, CA, USA).

**Table 4.** Chemical composition and C:N ratio of the tested bio-fertilizers.

| Nutrients | Bio-Fertilizers | | |
| | A (D = 25%) $kg\ t^{-1}$ | B (D = 75%) $kg\ t^{-1}$ | C (D = 45%) $kg\ t^{-1}$ |
|---|---|---|---|
| $N_t$ | 7.68 | 15.83 | 33.34 |
| $NH_4$-N | 1.05 | 3.15 | 1.89 |
| P | 12.20 | 6.86 | 34.09 |
| S | 49.75 | 0.00 | 49.75 |
| K | 35.52 | 58.49 | 42.69 |
| Ca | 107.09 | 41.62 | 93.74 |
| Mg | 15.41 | 8.81 | 9.50 |
| Na | 5.55 | 8.87 | 9.98 |
| Zn | 0.253 | 0.149 | 0.229 |
| Cu | 0.053 | 0.031 | 0.059 |
| Mn | 1.78 | 0.711 | 0.98 |
| Fe | 12.47 | 5.58 | 6.62 |
| $C:N_t$ | 14.2 | 20.9 | 5.9 |

Key: C—organic carbon (44%); $N_t$—total nitrogen; D—share of digestate dry weight in a biofertilizer.

**Table 5.** Rate of applied ammonium nitrogen in the tested bio-fertilizers, $kg\ ha^{-1}$.

| Bio-Fertilizer Type | Rate of Applied Bio-Fertilizer, $Kg\ ha^{-1}$ | | | |
| | 200 | 400 | 800 | 1600 |
|---|---|---|---|---|
| A | 1.5 | 3.1 | 6.1 | 12.3 |
| B | 3.2 | 6.3 | 12.6 | 25.3 |
| C | 6.7 | 13.3 | 26.7 | 53.3 |

2.4.2. Plant

The harvested samples tomato fruits were used for the determination of dry matter content (DM) and elements concentration. They were dried first (65 °C) for several days until a constant weight was obtained. Nitrogen concentration was determined using a standard macro-Kjeldahl procedure (Kjeltec Auto 1031 Analyzer, Foss Tecator, Hilleroed, Denmark). The plant materials for elements determination were mineralized at 600 °C. The obtained ash was then dissolved in 33% $HNO_3$. The phosphorus(P) concentration was measured by the vanadium-molybdenum method, using a Specord 250 at a wavelength of 436 nm (Analytik, Jena, Germany). The concentrations of K, Mg, Ca, Fe, Mn, Zn, Cu, Pb and Cd were determined, using FAAS (Atomic SpectrAA-55B, USA). For organic compound determination, the ripe fresh tomato fruit was minced with a juicer. Total soluble solids (TSS) of tomato juice were determined, using a digital refractometer (DR-103L) at 20 °C and results were reported as °Brix (EX). Total sugars (TS) content was determined by the anthrone method described by Yemm and Wills [43]. The extraction of carotenoids, using a mixture of acetone–hexane (4–6), was conducted according to procedure described by Nagata and Yamashita [44]. Lycopene and β-carotene concentration was estimated by measuring the absorbance of the acetone-hexane solution at 503 and 453 nm, respectively, on a UV-visible spectrophotometer.

*2.5. Statistical Analysis*

In order to assess the influence of treatments on tomato yield and quality, the two-way ANOVA was applied, evaluating the effects of individual research factors (year, fertilization treatments) and their interactions. The distribution of the data (normality) was checked, using the Shapiro-Wilk test. The homogeneity of variance was checked by the Bartlett test. Means were separated by honest significant difference (HSD), using Tukey's method, when the *F*-test indicated significant factorial effects at the level of $p < 0.05$. Principal component analysis (PCA) was applied for evaluation of the relationships between variables. The Tukey median is surrounded by a bag containing 50% of the data points. The bagplot visualizes the location, spread, correlation, skewness, and tails of data. The bagplot cover contains the inliers, and outside of the "fence" are outliers [45]. Statistica 13 software (TIBCO Software Inc., USA) was used for all statistical analyses [46].

## 3. Results

*3.1. Tomato Yield*

Total fruit yield of tomato significantly depended on the growing season and the fertilization treatment. However, no significant interaction was found for "year x fertilization treatment" (Supplementary Materials, Table S1). The highest yield was obtained in 2017 (173.7 t ha$^{-1}$). It was significantly higher in comparison with yields in other two years (Table S2). In 2016, it amounted to 95.4 t ha$^{-1}$ and in 2018 to 92.7 t ha$^{-1}$. The order of the tested bio-fertilizers, averaged over applied rates, was as follows: B (111.7) < C (119.2) < A (127.5 t ha$^{-1}$). The yields obtained were compared with yields from the Absolute Control (AC) plot, which was not fertilized, and with the Mineral Control plot (MC), representing the production standard (Tables 1 and 2). In general, no significant differences in total yields (TY) were found between the AC, MC, and plots with the tested bio-fertilizers, regardless of their type and applied rate. The highest yield was obtained on the MC plot, fertilized with the Yara Mila. The difference between the MC plot and B200, which yielded the lowest, was substantial, reaching 28.4 t ha$^{-1}$ (35.3%). Among the 12 studied treatments, the closest to the production standard, which yielded the highest, was the A400 plot (98.2%). In addition, the fruit yields responded differently to the rates of the tested bio-fertilizers. The highest fruit yield on the treatment with the A fertilizer was recorded on the plot with 400 kg ha$^{-1}$ (Figure 1).

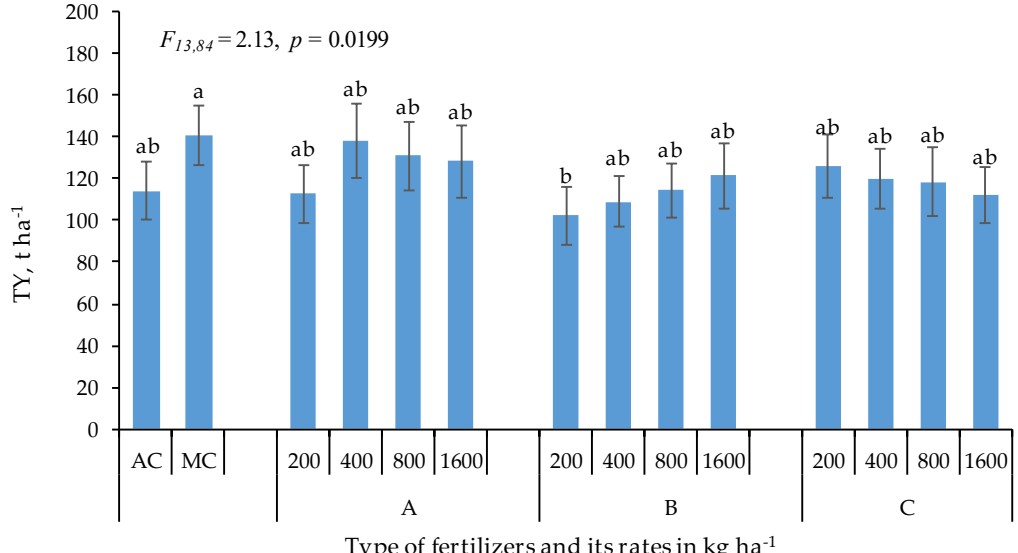

**Figure 1.** Effect of bio-fertilizer application on total yield (TY) of tomato (mean for 2016–2018). Letters indicate significant differences between treatments ($p < 0.05$). Hatched bars represent $2 \times$ standard error (*SE*) ranges. Key: A, B and C—bio-fertilizers; AC—absolute control; MC—mineral control.

A net increase, compared to the AC plot, and the plot with 200 kg ha$^{-1}$ (A200), was 21.0%, and 22.7%, respectively. Fertilizer rates greater than 400 kg ha$^{-1}$ resulted in an average tomato yield decrease by 5.3–7.2%. In plots fertilized with B fertilizer, the fruit yield increased progressively with the increasing fertilizer rate. The difference between plots fertilized with 200 and 1600 kg ha$^{-1}$ was only 18%. It is necessary to stress that TY on plots with 200 and 400 kg ha$^{-1}$ of applied fertilizer yielded less than that recorded on the AC plot. Plants fertilized with the C fertilizer yielded the highest on the C200 plot. Any higher rate of applied fertilizer resulted in a progressive yield reduction. The yield decrease, as recorded on the plot fertilized with 1600 kg ha$^{-1}$, reached 12%.

The commercial yield (CY) of tomato fruits in the study years contributed to 77.8%, 72.2%, and 62.3% of the TY, respectively in 2016, 2017, and 2018. The lower percentage share of the total commercial yield in 2018 resulted in a significant difference between 2016 and 2018 (Table S2), unlike the TY. There was no significant effect of the tested fertilizers on the marketable fruit yield. No significant interaction between the growing season and the fertilization treatments on the CY was found in the experiment. In 2016, the highest yield was obtained on the MC plot, and in 2017, 2018 on the A400 plot (Table S3). However, clear trends and differences were observed in the level of yields between individual treatments. Also, the value of the "*p*" statistic was close to the conventional critical value of 0.05. The relationships between the experimental treatments were very similar to those recorded for the TY (Figure 2 vs. Figure 1). The best yield-forming effect of the A bio-fertilizer was recorded on the plot treated with 400 kg ha$^{-1}$. It is also worth noting that, unlike TY, the average CY in the A400 variant was higher than that recorded in the MC plot. The difference was 6.5% (100.8 versus 94.7 t ha$^{-1}$). The increase in yield in this variant as compared to the AC was much higher, amounting to 27.7%. The tomato also yielded well on plot C, treated with 200 kg ha$^{-1}$. Compared to AC, the yield increase in the C200 variant was 15.9%. However, any further rate of this fertilizer increase resulted in a progressive yield decline. A reverse trend was observed on the main plot fertilized with the B bio-fertilizer. The lowest rates of this fertilizer resulted in a yield depression as compared to AC. The relative drop for B200 was 6%. The yield gap was not overcome until the application of 800 kg ha$^{-1}$ of this fertilizer.

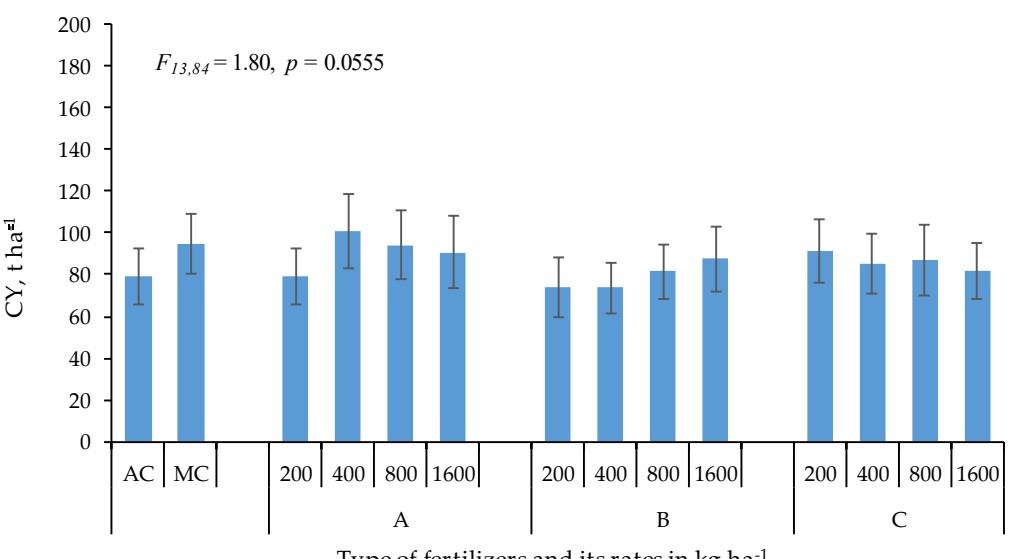

**Figure 2.** Effect of bio-fertilizer application on commercial yield (CY) of tomato (mean for 2016–2018). Hatched bars represent 2 × standard error (*SE*) ranges. Key: A, B and C—bio-fertilizers; AC—absolute control; MC—mineral control.

### 3.2. Yield Components

The growing season was the main factor determining the weight of fruits and number of fruits per m$^2$ (Table S1). For the TY, the fresh weight of a single fruit (TFW) in the subsequent years of the study was 157.9, 158.8 and 174.7 g, respectively (Table S2). After fruits selecting for commercial purposes, the weight of a single fruit (CFW), as expected, increased and amounted to 172.5, 161.0 and 180.8 g, respectively. Total fruit number (TFN) ranged from 53.2 to 109.4 per m$^{-2}$, depending on the year. For the CY, the fruit number (CFN) ranged from 31.9 to 77.9 per m$^2$, depending on the year. The fertilization factor had no significant influence on the fruit weight, both in the total and commercial yield. This factor, on the other hand, affected the TFN. The highest TFN was recorded on the plot fertilized with the mineral fertilizer, and was significantly lowest in the B200 variant. A similar trend was obtained for the number of fruits constituting the commercial yield. Both characteristics on the MC plot were by 18.9% and 17.8% higher as compared to the AC plot. When analyzing the effect of individual bio-fertilizers on the TFN, it was found that the optimal rate for the A fertilizer was 400 and 800 kg ha$^{-1}$. For the plots fertilized with the B fertilizer, it was the highest on the plot with 1600 kg ha$^{-1}$ and 400 kg ha$^{-1}$ for the C plots. The relative decrease in TFN as related to the MC was −5.2%, −1.6%, −12.6%, and −9.1%, respectively. The optimal rate for the maximum CFN was 400 and 800 kg ha$^{-1}$ for the A, 1600 kg ha$^{-1}$ for the B, and 400 kg ha$^{-1}$ for the C fertilizers, respectively. The net increase in CFN, as compared to the MC, was recorded on the A plot (+2.6% and 1.4%), and a substantial decrease on the B and C plots (−8 and −7%, respectively) was recorded (Table 6).

**Table 6.** Effect of bio-fertilizer application on weight and numbers of tomato fruits (mean ± standard error).

| Fertilizer/*F* Ratio | Rate Kg ha$^{-1}$ | TFW g | CFW G | TFN No. m$^2$ | CFN No. m$^2$ |
|---|---|---|---|---|---|
| AC |  | 160.1 ± 4.72 | 170.5 ± 5.74 | 71.9 ± 8.76 [ab] | 47.5 ± 7.20 |
| MC |  | 166.4 ± 6.61 | 172.8 ± 4.90 | 85.5 ± 8.76 [a] | 56.0 ± 7.21 |
| A | 200 | 160.9 ± 3.90 | 172.6 ± 4.53 | 70.8 ± 9.05 [ab] | 47.1 ± 7.49 |
|  | 400 | 170.6 ± 5.32 | 176.7 ± 5.18 | 81.1 ± 10.41 [ab] | 57.5 ± 8.21 |
|  | 800 | 155.5 ± 7.57 | 168.8 ± 3.69 | 84.1 ± 10.14 [ab] | 56.8 ± 8.02 |
|  | 1600 | 170.1 ± 5.47 | 171.6 ± 5.92 | 76.8 ± 11.20 [ab] | 53.7 ± 8.99 |
| B | 200 | 161.3 ± 5.22 | 169.4 ± 4.93 | 64.4 ± 9.30 [b] | 45.1 ± 7.01 |
|  | 400 | 162.5 ± 3.78 | 167.6 ± 4.08 | 67.7 ± 8.10 [ab] | 44.8 ± 6.50 |
|  | 800 | 167.7 ± 8.02 | 172.5 ± 9.52 | 69.5 ± 8.49 [ab] | 48.4 ± 6.68 |
|  | 1600 | 164.8 ± 5.70 | 172.0 ± 5.22 | 74.7 ± 9.90 [ab] | 51.5 ± 7.54 |
| C | 200 | 169.1 ± 5.90 | 179.5 ± 7.44 | 75.8 ± 10.28 [ab] | 52.2 ± 7.83 |
|  | 400 | 156.5 ± 4.39 | 161.5 ± 4.60 | 77.7 ± 9.89 [ab] | 53.7 ± 7.79 |
|  | 800 | 164.9 ± 4.96 | 171.2 ± 5.96 | 73.1 ± 11.28 [ab] | 52.1 ± 8.69 |
|  | 1600 | 162.9 ± 6.24 | 173.9 ± 6.30 | 70.6 ± 9.57 [ab] | 48.7 ± 7.14 |
| $F_{13,84}$ |  | n.s. | n.s. | 1.87 * | n.s. |

* significant at $p < 0.05$; n.s.—non significant; means within a column followed by the same letter indicate a lack of significant difference between the treatments. Key: A, B and C—bio-fertilizers; AC—absolute control; MC—mineral control, TFW—total weight of one fruit, CFW—commercial weight of one fruit, TFN—total number of fruits per m$^2$, CFN—commercial number of fruits per m$^2$.

### 3.3. Chemical Quality Parameters

The growing season significantly affected only one of the tomato fruit characteristics, i.e., tomato total extract (EX). In two years, i.e., 2016, 2017, its value was significantly higher than in 2018 (Table S2). The growing season, however, did not affect the other investigated tomato characteristics. On average, the EX content for A, B and C fertilizers, was at the level of 44.4, 45.1 and 45.5 g kg$^{-1}$, respectively. The EX content on the AC and MC was on the same level, amounting to 44.9 g kg$^{-1}$. The highest EX content was obtained on plots

fertilized with the C fertilizer at a rate of 800 kg ha$^{-1}$ (46.4 g kg$^{-1}$). The lowest EX content was recorded in fruits harvested from plots fertilized with the A fertilizer at rates of 400 and 1600 kg ha$^{-1}$, and with B fertilizer at the rate of 1600 kg ha$^{-1}$ (Table 7).

**Table 7.** Effect of bio-fertilizer application on content of dry matter and organic compounds (mean ± standard error).

| Fertilizer/*F* Ratio | Rate, kg ha$^{-1}$ | EX g kg$^{-1}$ FM | TS g kg$^{-1}$ FM | CRD mg kg$^{-1}$ FM | LCP mg kg$^{-1}$ FM | DM g kg$^{-1}$ FM |
|---|---|---|---|---|---|---|
| AC | | 44.9 ± 0.45 [ab] | 33.5 ± 0.88 [ab] | 317.8 ± 9.5 [b] | 226.4 ± 7.2 [bc] | 58.2 ± 0.84 [c] |
| MC | | 44.9 ± 0.58 [ab] | 37.2 ± 1.34 [a] | 371.0 ± 12.0 [ab] | 252.5 ± 11.4 [abc] | 61.4 ± 0.39 [abc] |
| A | 200 | 44.7 ± 0.39 [ab] | 34.5 ± 1.06 [ab] | 411.4 ± 30.5 [a] | 299.0 ± 16.8 [a] | 62.1 ± 0.82 [abc] |
| | 400 | 44.1 ± 0.36 [b] | 34.4 ± 1.04 [ab] | 315.0 ± 21.4 [b] | 234.2 ± 13.6 [bc] | 58.9 ± 0.68 [bc] |
| | 800 | 44.6 ± 0.45 [ab] | 32.0 ± 2.34 [ab] | 317.8 ± 11.0 [b] | 232.5 ± 7.8 [bc] | 60.2 ± 0.83 [abc] |
| | 1600 | 44.4 ± 0.29 [b] | 32.2 ± 0.91 [ab] | 386.2 ± 25.2 [ab] | 258.7 ± 17.4 [abc] | 60.1 ± 0.64 [abc] |
| B | 200 | 46.0 ± 0.24 [ab] | 35.2 ± 0.56 [ab] | 349.0 ± 20.5 [ab] | 232.5 ± 15.2[bc] | 62.1 ± 0.50 [abc] |
| | 400 | 44.8 ± 0.33 [ab] | 34.6 ± 0.29 [ab] | 333.1 ± 11.6 [ab] | 241.0 ± 3.7 [abc] | 62.1 ± 0.59 [abc] |
| | 800 | 45.4 ± 0.64 [ab] | 35.6 ± 1.50 [ab] | 321.8 ± 13.1 [b] | 218.8 ± 6.9 [c] | 63.6 ± 0.35 [ab] |
| | 1600 | 44.4 ± 0.17 [b] | 30.0 ± 1.57 [b] | 358.4 ± 17.0 [ab] | 246.0 ± 5.5 [abc] | 63.9 ± 0.55 [a] |
| C | 200 | 45.5 ± 0.44 [ab] | 35.8 ± 0.72 [ab] | 411.4 ± 11.8 [a] | 282.5 ± 12.0 [ab] | 62.2 ± 0.55 [abc] |
| | 400 | 44.6 ± 0.26 [ab] | 34.2 ± 1.11 [ab] | 372.3 ± 11.5 [ab] | 262.7 ± 9.0 [abc] | 63.3 ± 1.89 [abc] |
| | 800 | 46.4 ± 0.44 [a] | 38.0 ± 0.92 [a] | 398.5 ± 10.4 [ab] | 272.7 ± 11.1 [abc] | 64.5 ± 2.08 [abc] |
| | 1600 | 45.3 ± 0.30 [ab] | 35.9 ± 0.97 [ab] | 335.5 ± 20.2 [ab] | 235.5 ± 12.9 [bc] | 62.6 ± 0.63 [abc] |
| $F_{13,84}$ | | 2.70 ** | 2.67 ** | 4.14 *** | 3.39 *** | 3.23 *** |

***, **, significant at *p* < 0.001, *p* < 0.01, respectively; means within a column followed by the same letter indicate a lack of significant difference between the treatments. Key: A, B and C—bio-fertilizers; AC—absolute control; MC—mineral control, EX—total extract, TS—total sugar, CRD—carotenoids, LCP—lycopene, DM—dry matter, FM—fresh matter.

The content of total sugar (TS) in fruits fertilized with mineral fertilizer was 37.2 g kg$^{-1}$, being by 11.1% higher as compared to that recorded on the AC. The highest TS, i.e., at the level of MC, was also found for the C800 plot (+2.1% vs. MC). Compared to these variants, TS in fruits harvested from the B1600 plot was significantly lower. On average, the effect of fertilizers on TS was as follows: A (33.3) < B (33.8) < C (36.0 g kg$^{-1}$). Tomato fruits fertilized with mineral fertilizer contained by 16.4% more carotenoids (CRD) as compared to the AC. The highest content of CRD was obtained after applying fertilizers A and C at the rate of 200 kg ha$^{-1}$. These contents were significantly higher than those obtained on the MC, and on the plot fertilized with A400, A800 and B800. On average, the effect of the tested bio-fertilizers on CRD was as follows: B (340) < A (358) < C (379 g kg$^{-1}$). Tomato fruits fertilized with Yara Mila contained by 11.5% more lycopne (LCP) as compared to the AC. The highest content was obtained on plots A200 and C200. The LCP content was greater than on the AC and on the A400, A800, B200, B800, C1600 plots. On average, the effect of the tested bio-fertilizers on LCP was as follows: B (235) < A (256) < C (263 g kg$^{-1}$). The lowest content of dry matter (DM) was obtained in tomato fruits harvested on the AC. In turn, the highest DM value was recorded in the C800. It was higher by 10.8% and 5%% with respect to the AC and MC, respectively. On average, the effect of the tested fertilizers on DM was as follows: A (60.3) < B (62.9) ≤ C (63.2 g kg$^{-1}$).

### 3.4. Nutrients and Heavy Metals

The content of most of the examined nutrients in tomato fruits depended on the growing season (Tables 8 and 9). The exceptions were nitrogen (N) and potassium (K), whose contents were driven by the interaction of years and rates of applied fertilizers (Tables S4 and S5). The effect of the applied fertilizer rate on the N content was specific for the studied bio-fertilizers. It is necessary to stress that the N content in tomato fruits grown on the AC was higher as compared to that recorded on the MC. The N content in tomato fruits on plots fertilized with the A fertilizer increased progressively up to its rate of 800 kg ha$^{-1}$. The highest increase was recorded on the A400 as compared to A200. It also exceeded the N content recorded for fruits from the MC. A similar trend was observed on

plots fertilized with the C fertilizer. For the B fertilizer, the highest average N content was obtained for the highest its rate. Regardless of the rate, the effect of the tested fertilizers on N content was as follows: A (31.1) < B (32.2) < C (34.0 g kg$^{-1}$). Among all studied treatments, including the MC, the highest content of N in tomato fruits was recorded for plants fertilized with the C fertilizer at the rate of 800 kg ha$^{-1}$. The N content was significantly higher as compared to the A200, A400, B200, B400 and MC plots. Contrary to N, the K content in the tomato fruits grown on the MC was significantly higher in comparison to the AC (Table 3).

**Table 8.** Effect of bio-fertilizer application on macronutrients content of tomato fruits in g kg$^{-1}$ DM (mean ± standard error).

| Fertilizer/F Ratio | Rate kg ha$^{-1}$ | N | P | K | Mg | S | Na |
|---|---|---|---|---|---|---|---|
| AC | | 32.2 ± 1.2 [abc] | 0.36 ± 0.03 | 37.4 ± 2.7 [c] | 2.05 ± 0.14 | 0.62 ± 0.06 | 0.35 ± 0.04 |
| MC | | 30.7 ± 0.7 [bcd] | 0.40 ± 0.01 | 40.7 ± 2.6 [ab] | 2.15 ± 0.07 | 0.69 ± 0.06 | 0.44 ± 0.06 |
| A | 200 | 27.8 ± 0.7 [d] | 0.43 ± 0.01 | 39.9 ± 1.9 [ab] | 2.14 ± 0.07 | 0.67 ± 0.05 | 0.46 ± 0.06 |
| | 400 | 31.9 ± 1.0 [bc] | 0.46 ± 0.02 | 38.5 ± 2.8 [bc] | 2.21 ± 0.05 | 0.66 ± 0.05 | 0.41 ± 0.04 |
| | 800 | 32.5 ± 0.6 [abc] | 0.40 ± 0.01 | 41.4 ± 1.9 [a] | 2.26 ± 0.05 | 0.62 ± 0.06 | 0.43 ± 0.05 |
| | 1600 | 32.3 ± 1.3 [abc] | 0.38 ± 0.03 | 41.5 ± 1.8 [a] | 2.30 ± 0.04 | 0.70 ± 0.05 | 0.43 ± 0.05 |
| B | 200 | 31.3 ± 1.2 [bcd] | 0.43 ± 0.03 | 39.5 ± 2.8 [abc] | 2.25 ± 0.05 | 0.73 ± 0.05 | 0.44 ± 0.04 |
| | 400 | 30.7 ± 0.5 [cd] | 0.44 ± 0.01 | 40.4 ± 2.3 [ab] | 2.18 ± 0.06 | 0.72 ± 0.05 | 0.40 ± 0.04 |
| | 800 | 32.3 ± 0.5 [abc] | 0.38 ± 0.02 | 40.2 ± 2.3 [ab] | 2.26 ± 0.05 | 0.67 ± 0.06 | 0.41 ± 0.04 |
| | 1600 | 34.6 ± 0.6 [abc] | 0.50 ± 0.06 | 40.9 ± 2.4 [ab] | 2.25 ± 0.04 | 0.69 ± 0.07 | 0.41 ± 0.03 |
| C | 200 | 34.8 ± 1.0 [ab] | 0.43 ± 0.01 | 40.5 ± 2.7 [ab] | 2.19 ± 0.11 | 0.75 ± 0.18 | 0.42 ± 0.03 |
| | 400 | 32.6 ± 1.0 [abc] | 0.44 ± 0.06 | 41.1 ± 2.2 [a] | 2.14 ± 0.06 | 0.74 ± 0.07 | 0.38 ± 0.05 |
| | 800 | 36.3 ± 0.8 [a] | 0.44 ± 0.01 | 41.3 ± 2.4 [a] | 2.27 ± 0.05 | 0.70 ± 0.08 | 0.46 ± 0.04 |
| | 1600 | 32.5 ± 1.2 [abc] | 0.46 ± 0.01 | 40.8 ± 2.5 [ab] | 2.30 ± 0.07 | 0.73 ± 0.06 | 0.42 ± 0.04 |
| $F_{13,84}$ | | 5.91 *** | n.s. | 5.56 *** | n.s. | n.s. | n.s. |

***, significant at $p < 0.001$; n.s.—non significant; means within a column followed by the same letter indicate a lack of significant difference between the treatments. Key: A, B and C—bio-fertilizers; AC—absolute control; MC—mineral control.

**Table 9.** Effect of bio-fertilizer application on micronutrients, lead and cadmium content of tomato fruits in mg kg$^{-1}$ DM (mean ± standard error).

| Fertilizer/F Ratio | Rate, kg ha$^{-1}$ | Zn | Cu | Mn | Fe | Pb | Cd |
|---|---|---|---|---|---|---|---|
| Control | | 1.90 ± 0.11 | 0.79 ± 0.06 | 1.60 ± 0.10 | 5.52 ± 0.83 | 0.49 ± 0.15 | 0.029 ± 0.002 |
| YaraMila | | 1.97 ± 0.10 | 0.79 ± 0.03 | 1.67 ± 0.09 | 5.18 ± 0.45 | 0.31 ± 0.08 | 0.026 ± 0.002 |
| A | 200 | 2.12 ± 0.08 | 0.87 ± 0.05 | 1.75 ± 0.09 | 5.77 ± 0.39 | 0.37 ± 0.09 | 0.028 ± 0.002 |
| | 400 | 1.97 ± 0.08 | 0.83 ± 0.03 | 1.65 ± 0.09 | 5.28 ± 0.31 | 0.37 ± 0.11 | 0.027 ± 0.002 |
| | 800 | 2.04 ± 0.10 | 0.86 ± 0.03 | 1.76 ± 0.08 | 5.61 ± 0.50 | 0.35 ± 0.11 | 0.027 ± 0.001 |
| | 1600 | 2.07 ± 0.08 | 0.87 ± 0.03 | 1.73 ± 0.11 | 5.93 ± 0.53 | 0.28 ± 0.04 | 0.028 ± 0.002 |
| B | 200 | 2.03 ± 0.08 | 0.88 ± 0.04 | 1.71 ± 0.08 | 5.51 ± 0.39 | 0.61 ± 0.26 | 0.026 ± 0.001 |
| | 400 | 1.99 ± 0.11 | 0.89 ± 0.10 | 1.68 ± 0.08 | 5.61 ± 0.37 | 0.46 ± 0.23 | 0.027 ± 0.001 |
| | 800 | 2.03 ± 0.11 | 0.87 ± 0.05 | 1.79 ± 0.10 | 6.35 ± 0.55 | 0.72 ± 0.48 | 0.027 ± 0.003 |
| | 1600 | 1.99 ± 0.10 | 0.92 ± 0.04 | 1.61 ± 0.06 | 5.88 ± 0.76 | 0.37 ± 0.09 | 0.028 ± 0.001 |
| C | 200 | 1.98 ± 0.14 | 0.76 ± 0.04 | 1.96 ± 0.20 | 5.92 ± 0.29 | 0.24 ± 0.04 | 0.025 ± 0.003 |
| | 400 | 1.76 ± 0.18 | 0.67 ± 0.06 | 1.66 ± 0.12 | 5.65 ± 0.59 | 0.23 ± 0.03 | 0.026 ± 0.003 |
| | 800 | 2.71 ± 0.24 | 0.87 ± 0.06 | 2.89 ± 0.28 | 6.63 ± 0.71 | 0.51 ± 0.24 | 0.032 ± 0.003 |
| | 1600 | 1.97 ± 0.10 | 0.85 ± 0.02 | 1.73 ± 0.06 | 6.49 ± 0.90 | 0.36 ± 0.10 | 0.028 ± 0.002 |
| $F_{13,84}$ | | n.s. | n.s. | n.s. | n.s. | n.s. | n.s. |

n.s.—non significant. Key: A, B and C—bio-fertilizers; AC—absolute control; MC—mineral control.

The only exception were variants A800, A1600, and B200 and B400, for which the level of the K content in fruits was significantly higher compared to the MC. For the A and B fertilizers, there was a clear trend towards the K content increase in accordance with the

amount of applied fertilizer. For the C fertilizer, the highest K content was obtained in the C800. On average, the effect of the tested bio-fertilizers on the K content was as follows: B (40.3) $\leq$ A (40.4) $\leq$ C (40.9 g kg$^{-1}$). Fertilization had no significant effect on the content of micronutrients and toxic elements, i.e., cadmium and lead (Table S3). There were also no specific trends in changes in the content of these elements as a result of the applied doses and types of fertilizers (Table 4).

### 3.5. Relationships between Features

Correlation analysis and principal component analysis (PCA) were used to determine the relationships between the examined tomato characteristics. These relationships were analyzed in terms of variability caused by the fertilization factor, i.e., regardless of the years. The values of the correlation coefficients can be seen in the supplemental materials (Tables S6 and S7). The results of the PCA procedure were visualized in biplots. Based on the PCA, three main components, representing the yield, yield components and the content of organic compounds, accounted for 79.0% of the total variance. The first principal component (PC1) explained 40.5% of the total variability, and the next two components (PC2 and PC3), respectively 23.9% and 14.6% of the total variance (Figure 3). PC1 consisted of variables related to the tomato yield (TY and CY), as well as the TFN, CFN. The loading exerted by LCP and CRD influenced PC2. As shown in Figure 3a, the CY was significantly related to the total yield. High values of the correlation coefficient (r > 0.90) were also obtained for relationships between TY and TFN, and between CY and TY, and CY and CFN. The TY showed negative relationships with the content of extract (EX), and dry matter (DM). The content of LCP and CRP was positively correlated with the content of TS, and secondly with the content of DM, and the weight of a single fruit. The results presented on the PC1-PC3 biplot show a very similar pattern to that of the PC1-PC2 one. The greatest difference concerned the correlation of the DM with other characteristics, in particular with the EX (Figure 3b). The fertilization treatments modified the values of the investigated parameters, as demonstrated by the both PCA biplots. On the PC1−PC2 biblot axes, treatments A1600, B1600, C1600, B800 and C400 were grouped closest to the Tukey median (in the bagplot). The other treatments, including the Absolute Control (AC) and Yara Mila plots (MC), were located in the bagplot cover region. The B200 plot was on the verge of the bagplot, in the opposite direction to A400 and MC. On the second biplot, the B200 treatment was separated by a significant distance from the Tukey median, and it is located in the opposite direction to A400 and A800 treatments (Figure 3b).

For the marketable yield (CY) and variables, representing mineral composition of tomato fruits, two principal components are obtained (Figure 4). Together, they account for 52.8% of the total variance. The PC1 consisted of variables, representing mainly Zn, Mn, Fe and Mg. The PC2 was composed mainly by the variables representing the contents of K, Ca and Pb. The analysis of the PCA biplot axes revealed that the AC plot was separated by a significant distance from the Tukey median, and it is located between the axes for variables representing contents of N and K. On the opposite side are located treatments with the highest amount of applied fertilizers, i.e., A1600, B1600 and C1600. All are located in the bagplot. The outliers are the B800, C200, C400, C800, and the AC plot. The A400 and C800 plots are close to the Mn and Fe lines. Both nutrients were significantly correlated with each other, but without an impact on the CY yield (Figure 4).

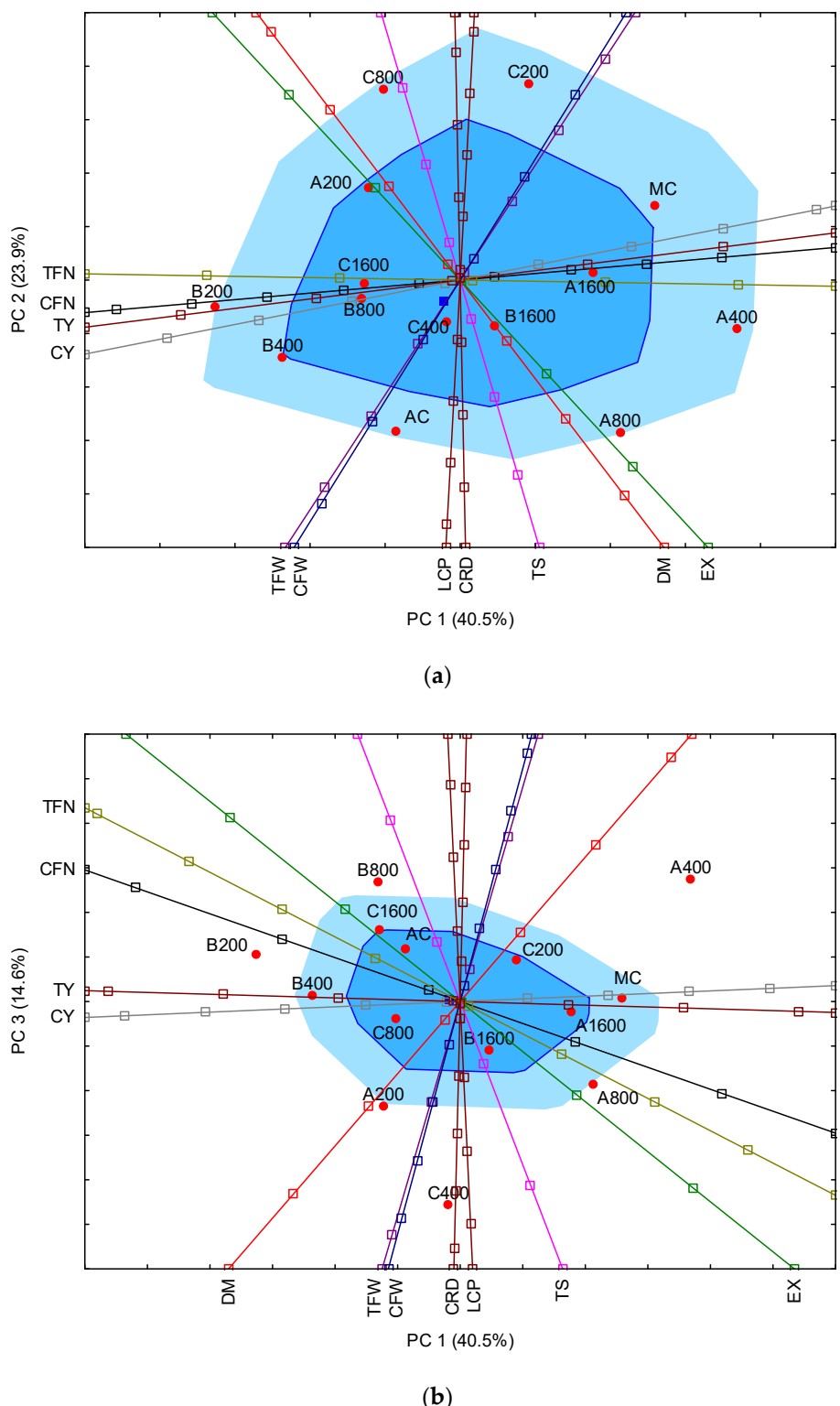

**Figure 3.** (**a**,**b**) Principal component analysis (PCA) biplot of the tomato yield, yield components and content of organic compounds. The dark blue square denotes the Tukey median, the blue square is the bagplot, the light blue square is the bagplot cover. Key: A, B and C—bio-fertilizers; AC—absolute control; MC—mineral control, TFW—total weight of one fruit, CFW—commercial weight of one fruit, TFN—total number of fruits per m², CFN—commercial number of fruits per m², EX—total extract, TS—total sugar, CRD—carotenoids, LCP—lycopene, DM—dry matter.

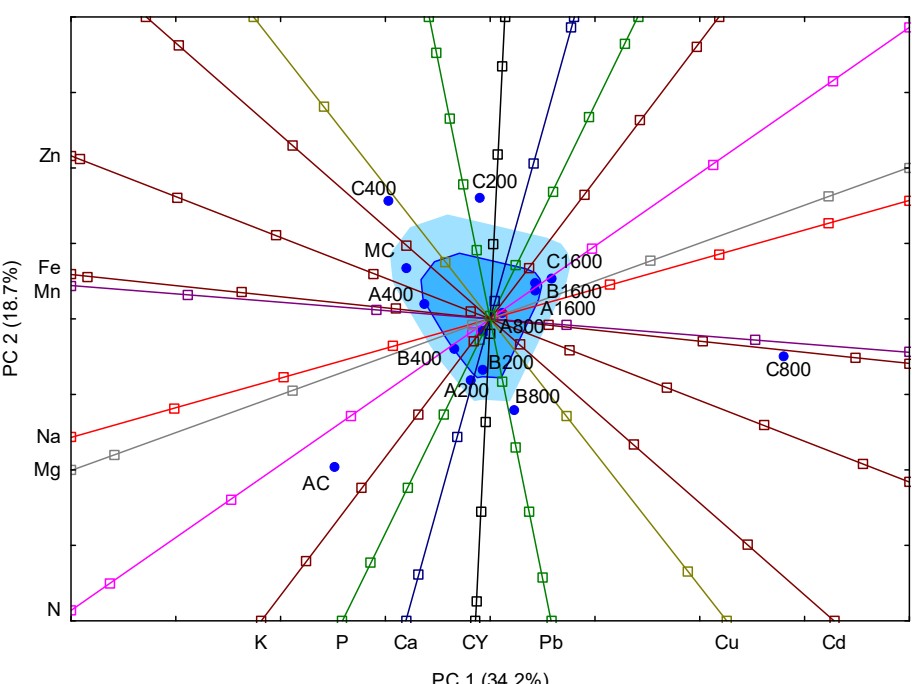

**Figure 4.** Principal component analysis (PCA) biplot of the tomato commercial yield (CY) and content in fruits of elements. The dark blue square denotes the Tukey median, the blue square is the bagplot, the light blue square is the bagplot cover. Key: A, B and C—bio-fertilizers; AC—absolute control; MC—mineral control.

## 4. Discussion

Application of bio-fertilizers based on recycled materials is, in general, considered, as an eco-friendly solution, leading to a decrease of non-renewable resources use [47]. An application of bio-fertilizers as nutrient carriers to vegetables creates, however, both an economic risk for a farmer, and a health risk for a consumer. The sound use of this type of nutrient carries requires at least four main end-effects to be taken into account,: (i) yield of edible parts as compared to that recorded for a standard mineral fertilizer, (ii) quality of edible parts, including both the content of phytochemicals and minerals, (iii) threat of the contamination of edible parts by heavy metals, (iv) soil depletion with available nutrients [21,25,48].

The three year study of applied bio-fertilizers, with tomato as a test crop, clearly showed that weather conditions during the growing season were the factor significantly affecting fruit yield, irrespective of the type and rate of applied bio-fertilizers (Figure 1). In the wet 2017, both total (TY) and commercial (CY) yields were two-fold higher on the AC plot as compared to the relatively normal 2016, and the hot 2018. The commercial fruit yields of ground tomatoes on the AC plot of 53 t ha$^{-1}$ in 2018, 60 t ha$^{-1}$ in 2016, and 120 t ha$^{-1}$ in 2017, clearly stresses two facts. The first indicates a high natural productivity of the soil under study. The commercial yield of varieties of ground tomato, as reported by Korzeniowska [49], ranges from 20 to 60 t ha$^{-1}$. A higher yield can be obtained only by applying irrigation [50]. In our study, the fruit yields harvested on the plot with the standard rate of mineral fertilizer of 100 kg N ha$^{-1}$ were 91.6, 133.5, and 59 t ha$^{-1}$ in 2016, 2017, and 2018, respectively. The average total yields, harvested on plots fertilized with the tested bio-fertilizers, were 73.8, 125.2, and 58 t ha$^{-1}$, in respective years. The main reasons for the yield drop in 2018 were a shortage of precipitation and concomitant, elevated temperatures in August. It is well documented that a higher temperature following flowering results in a poor set of tomato fruits, subsequently leading to a significant fruit yield decrease [51]. Ronga et al. [52] studying the effect of digestate on tomato production in organic farming, have achieved 47 t ha$^{-1}$ of fruits on an unfertilized plot, and 59 t ha$^{-1}$

on a plot with digestate. The second point indicates the dominant role of water supply during fruiting, yield development and ripening of tomato. Unit productivity of rainfall water, in spite of huge differences in fruit yields between years, was much lower. As shown in Table 10, this parameter increased in the order (averaged over years for CY): AC < Bio-fertilizers < MC (257.3 ± 27.5; 284 ± 30.2; 315.7 ± 67.8 kg fruits $mm^{-1}$ rainfall water). The year to year variability of water productivity was 21% for MC, and was 2-fold lower for other treatments. This small variability suggests a stabilization of water productivity by application of bio-fertilizers. The highest values, as recorded for the MC plot corroborate the sensitivity of tomato both to water and nutrient supply [53].

**Table 10.** Unit productivity of rainfall water during the tomato growing season, kg fruits per 1 mm.

| Year | Yield Category | Absolute Control | Mineral Control | Bio-Fertilizer |
|------|----------------|------------------|-----------------|----------------|
| 2016 | Total | 349 | 469 | 402 |
|      | Commercial | 268 | 387 | 312 |
| 2017 | Total | 385 | 450 | 397 |
|      | Commercial | 278 | 308 | 288 |
| 2018 | Total | 397 | 493 | 387 |
|      | Commercial | 226 | 252 | 252 |

Our study clearly showed that the tested bio-fertilizers showed a highly differentiated impact on the total fruit yield (Figure 1). Tomato plants fertilized with 400 kg $ha^{-1}$ of the A fertilizer, produced, among the studied treatments, the highest yield, both total and commercial. The total yield on this plot was only slightly lower, but at the same time, the commercial yield was significantly above (106.5%) that obtained on the plot fertilized with a mineral fertilizer at a dose of 100 kg N $ha^{-1}$. This N dose is rather low as compared to that applied in practice and in experiments [53,54]. The rates of the A fertilizer above 400 kg $ha^{-1}$ resulted in the yield drop. Yield trends for the other two bio-fertilizers showed quite opposite trends in response to their increasing rates. The fruit yield on the plot fertilized with the B fertilizer increased in accordance with the increase of its rate. It is necessary to stress that on plots treated with 200 and 400 kg $ha^{-1}$, both total and commercial yields were lower in comparison to the AC, indicating a yield depression. On plots treated with the C bio-fertilizer, the fruit yield was the highest on the plot with 200 kg $ha^{-1}$/Then it declined in response to higher rates of applied C bio-fertilizer. (Figure 2). The apparently contradictory results of the impact of the tested bio-fertilizers on tomato fruit yield presented above show, however, a logical correctness. The yielding effect of the applied bio-fertilizers based on digestate and biomass ash to a considerable extent followed the $C:N_t$ ratio, which increased in the order: C ($\approx$6), A ($\approx$13), and B ($\approx$19). The order of TYs and CYs were, respectively as follows: 119.2/86.4, 127.5/91.3; 111.7/79.2 t $ha^{-1}$. The orders obtained clearly show that the highest yields were recorded on plots fertilized with the A bio-fertilizer. The observed effect can be indirectly explained by the amount of N supplied to plants. The number of fruits and the content of N in the ripe tomato fruits on the A400 plot, which yielded the highest (CY) were, in general, at the same level as recorded on the MC plot, indirectly indicating a good supply of N both to plants during fruiting, as well as during fruit development and ripening. A balanced N supply to plants during the critical period of yield development is crucial for yield in tomato [55]. The best effect of the A fertilizer can be explained by $C:N_t$ ratio, which was at the level of high-quality farmyard manure.

The observed opposite directions in both fruit yield, and N content, as revealed on plots fertilized with increasing rates of B and C fertilizers, can also be explained by the $C:N_t$ ratio. The main reason for the observed trends on plots treated with the B fertilizer was a shortage of N supply. The fruit yield is much less affected by a shortage of N, as compared to leaves and stems, and leads to yield drop [56]. An excess of N supply was recorded on plots fertilized with the C fertilizer. Excessive rates of fertilizer N, as reported by Warner at al. [57], result in an increased number of green fruits at harvest.

In our case, the observed trends can be explained by an analysis of the two main yield formic components, i.e., fruit density (total, commercial fruit-number, TFN, CFN, and fruit weight (TFW, CFW). The second yield component was neutral with respect to its impact on fruit yield ($R^2 \approx 0.15$; n.s.). This type of relationship also indicates that tomato yields were driven by one factor, i.e., fruit density. This assumption was fully corroborated in this study, as shown by equations presented below:

$$TY = 1.729TFN - 8.275 \text{ for } n = 11, R^2 = 0.90 \text{ and } p < 0.001 \qquad (1)$$

$$CY = 1.838CFN - 8.149 \text{ for } n = 11, R^2 = 0.93 \text{ and } p < 0.001 \qquad (2)$$

The next important question refers to the relationships between yield and the basic set of quality indicators of tomato yield, such as content of dry matter (DM), total extract (EX), total sugars (TS), carotenoids (CRD), and lycopene (LCP). The study showed that all these fruit characteristics responded significantly to the studied bio-fertilizers and their rates (Table 7). At the same time, all these qualitative characteristics were not significantly correlated with the tomato yield. The lack of these relationships could be a clear indication that tomato plants fertilized with bio-fertilizers based on digestate and biomass ashes, were well supplied with nutrients, with no negative impact on the fruit quality [58]. Some of the studied treatments showed, however, specific relationships with tomato fruit's qualitative characteristics. Two pairs of tomato fruit characteristics can be considered. The first one is total sugar (TS) and total extract (EX) contents. Both features were correlated with each other ($r = 0.70$; $p < 0.001$), clearly stressing that the yield obtained in the A400 plot was not affected by these characteristics (Figure 3). However, the highest rates of the B fertilizer resulted in a significant drop of both characteristics, as compared to the effect of the mineral fertilizer. The observed trend suggests a shortage of N supply to the ripening fruits [55].

The second significant relationship was observed between the contents of carotenoids (CRD) and lycopene (LCP) ($r = 0.94$; $p < 0.001$). This type of relationship is well documented because lycopene is the key component of total carotenoids in tomato fruits. In the studied case, the content of these two characteristics increased in the order B < A < C, i.e., in accordance with the potential N supply, as determined by the $C:N_t$ ratio in the applied bio-fertilizers. The share of lycopene in the total carotenoids was 69%, 72%, and 69%, respectively. These values are, however, much below the referenced data [24]. The highest values of both characteristics were recorded on the A200 plot, which yielded less by 20% with respect to the MC.

The content of minerals is an important characteristic of tomato fruits due to their contribution to the daily human diet [24,48]. The contents of most nutrients in tomato fruits were within the literature ranges [59]. The contents of N and K were driven by the interaction of the applied bio-fertilizer, including its rates, and the course of the weather during the tomato growing season (Table S3). The N content was a nutritional factor, defining the fruit yield obtained on plots with the highest applied rates of the tested bio-fertilizers. These plots are located in Tukey median (A, B) and bagplot (C) (Figure 4). The importance of N and K for tomato, as yielding factors, is clearly demonstrated in this study by the localization of the AC plot in the bagplot cover, stressing their deficiency for plants. These sets of data indirectly stress that the supply of K and N to tomato plants, was deficient, but only for those on the AC plot.

The K content in tomato fruits in 2018 was, averaged over treatments, significantly higher compared to both previous years, but especially to 2017. In this particular year, the K content reached 32.0 g kg$^{-1}$ DW, i.e., below the lower range [51], which is fixed at 36 g kg$^{-1}$ DW. In contrast, in 2018, the K content reached 48.2 g kg$^{-1}$ DW, i.e., close to the upper level of the range, which is 48.33 g kg$^{-1}$ DW. A good supply of K to tomato fruits, taking into account a deep seasonal variability, stresses its important role in water management [60]. A detailed analysis showed that the lowest K content in 2017 was 28.5 g kg$^{-1}$ DW, but harvested yield reached 199.5 t ha$^{-1}$. In 2018, the highest K content was almost 2-fold higher, amounting to 50.6 g kg$^{-1}$ DW. At the same time, tomato yield, as recorded on the

C200 plot was 97.1 t ha$^{-1}$. These two sets of tomato fruit characteristics clearly corroborate the opinion that some nutrients, in this case K, undergo the phenomenon known as the *dilution effect* [61]. It is necessary to stress that in the dry weather conditions, as dominated in 2018, the content of K for most plots treated with bio-fertilizers was at the level recorded for the mineral fertilizer. This phenomenon can be explained directly by the fact that the highest contents of N and K were recorded on plots treated with C fertilizer. The main reason for the high K content can be explained by the C:N$_t$ ratio in the applied bio-fertilizers. The first product of organic N mineralization is ammonium, which in the soil undergoes solution transformation into ammonium ion (NH$_4^+$) [62]. As a rule, this cation can exchange potassium ions (K$^+$) from the cation exchange complex [63]. The content of K did not show a close relationship with the content of lycopene as frequently reported in literature [64].

Magnesium is considered as one of the most sensitive nutrient, undergoing a *dilution effect* in response to the increased yield of vegetables and fruits [61]. An enrichment of digestate with magnesium is an useful option to get both a higher yield and better quality of vegetables, for example kohlrabi [65]. This phenomenon was not observed in our study. The Mg content, with the exception of the AC plot, was above the upper level of the standard range 2.07 g kg$^{-1}$ DW [59]. Its content was positively correlated with the K content (Table S7). The contents of all other nutrients were in the Marles' ranges [59].

Vegetables and fruits fertilized with organic fertilizers from recycled agricultural wastes are potentially threatened by harmful elements, such as Cd and Pb [25]. The study showed that an increasing content of Pb resulted in a CY decline (Table S7). The recorded Pb content in tomato fruits was, in general, low, ranging from 0,23 to 0,72 mg kg$^{-1}$ DW. Even the highest value was below the threshold standard for lead, amounting to 0.1 mg kg$^{-1}$ FW [29]. The same level of Pb content in tomato fruits was reported by Li [31]. The negative impact of the Pb content on tomato commercial yield was revealed in fruits mostly from low-yielding plots, such as AC, B200, B800, and C800. The content of Cd was extremely low, ranging from 0.025 to 0.032 mg kg$^{-1}$ DW, i.e., below, after recalculation, the threshold standard of 0.1 mg kg$^{-1}$ FW [29]. The obtained Cd content was 10-fold lower as compared to data reported by [31]. The positive relationships of the Cd content with the contents of Zn and Mn result from the same set of membrane transporters responsible for their movement in the phloem towards the developing fruit [66]. The excessive content of available heavy metals can be efficiently controlled by an adequate content of plant available K and Zn in the soil. In our study, the content of most nutrients, including K, Ca, Mg, and Zn in the soil, was in the good or high classes. Thus, the threat of tomato fruit contamination with heavy metals was low.

## 5. Conclusions

The production effect of the tested bio-fertilizers as compared to the standard mineral fertilizer was positive. The key characteristic of a particular bio-fertilizer, decisive for fruit yield, was C:N$_t$ ratio, affecting the supply of nitrogen to tomato plants. An indirect indicator of N supply was the number of fruits per m$^2$ (TFN, CFN) which was a single yield predictor. Based on the C:N$_t$ ratio, the average yield of fruits followed the studied bio-fertilizers in the order: B < C < A. The maximum yield on plots treated with the B fertilizer was achieved on the plot with its maximum rate of 1.6 t ha$^{-1}$. The main reason for this was a C:N$_t$ ratio of 21:1. The lowest rates of this fertilizer resulted in a yield depression, as compared to that on the Absolute Control plot. A C:N$_t$ ratio of 6:1 in the C fertilizer resulted in a rate of 0.2 t ha$^{-1}$ which was high enough to achieve the highest yield for this set of plots (−3.4% with respect to the mineral fertilizer standard). It is necessary to stress that rates of C fertilizer above 0.2 t ha$^{-1}$ resulted in the yield drop. The best production effect was recorded for the A fertilizer, which applied at a rate of 0.4 t ha$^{-1}$, resulted in a 6.5% higher yield as compared to the mineral standard. The contents of total extract and phytochemicals showed a significant response to the tested bio-fertilizers, but at the same time their contents were well balanced with the fruit yield. Among the examined

macro and micronutrients, only nitrogen and potassium showed a year-to-year variability, being, however, in balance with fruit yield. The contents of all other nutrients were in ranges reported in scientific reports. The content of lead, in spite of its negative impact on fruit yield, was below the threshold value of 0.1 mg kg$^{-1}$ FM. The presented work clearly stresses that bio-fertilizers based on digestate and biomass ash can replace mineral fertilizer, provided that the dosage is well-fixed and the soil has a high level of fertility.

**Supplementary Materials:** The following are available online at https://www.mdpi.com/article/10.3390/agronomy11091716/s1, Table S1. The results of ANOVA analysis: yield, yield components and quality of tomato fruits (*F* ratios); Table S2. Effect of the year on tomato yield, yield components and quality of tomato fruits; Table S3. Effect of interaction between year and treatments on total yield (TY), commercial yield of tomato (CY) and content of nitrogen and potassium in tomato fruits; Table S4. The results of ANOVA analysis: content of nutrients, lead and cadmium (*F* ratios); Table S5. Effect of the year on nutrients, lead and cadmium content in tomato fruits; Table S6. Matrix of Pearson's correlation coefficients between tomato yield, yield components and quality parameters (n = 11); Table S7. Matrix of Pearson's correlation coefficients between commercial yield of tomato (CY) and content of elements in fruits (n = 11).

**Author Contributions:** Conceptualization, K.P.-C. and W.G.; methodology, P.B. and W.G.; validation, K.P.-C. and P.B.; formal analysis, K.P.-C. and P.B.; investigation, K.P.-C. and T.S.; resources, K.P.-C. and T.S.; data curation, K.P.-C. and P.B.; writing—original draft preparation, K.P.-C. and P.B.; writing—review and editing, K.P.-C. and W.G.; visualization, P.B.; supervision, W.G.; project administration, K.P.-C. and T.S. All authors have read and agreed to the published version of the manuscript.

**Funding:** The study was financed under the program of the Ministry of Science and Higher Education 'Regional Initiative of Excellence' in the years 2019–2022; Project No. 005/RID/2018/2019.

**Institutional Review Board Statement:** Not applicable.

**Informed Consent Statement:** Informed consent was obtained from all subjects involved in the study.

**Conflicts of Interest:** The authors declare no conflict of interest.

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
