# Peer review of "Bio-Fertilizers Based on Digestate and Biomass Ash as an Alternative to Commercial Fertilizers—The Case of Tomato"

_agronomy, doi:10.3390/agronomy11091716_

Round 1

Reviewer 1 Report

Taking tomato as an example, this manuscript explored the feasibility of replacing commercial fertilizer with bio-fertilizer based on digestate and biomass ash, and studied the effects of bio-fertilizer on tomato yield, yield components, contents of organic compounds and minerals in fruits compared with standard inorganic fertilizer. The research content of this manuscript has reference significance for the actual production of plants and environmental protection. Using biological fertilizer instead of common commercial fertilizer can save resources for production.

Although this manuscript can provide a theoretical basis for production, the author's writing needs to be improved. The introduction part lacks the description of the research background in this field, and more emphasis is on the repeated elaboration of the concept and significance of biological fertilizer. And there are some problems in article format and other aspects that need to be improved, as follows:

  1. It is difficult to understand that the text notes below Tables 1 and 2 are connected with the same contents in the table, and the units in the table are also confused, which is difficult to correspond to the chemical content. It is suggested to make the revision more concise.
  2. The weather conditions in the materials and methods do not mention the source of information, it is suggested to add information sources.
  3. Please explain why each of the three types of biological fertilizer to be tested is selected according to the four quantities of 200, 400, 800 and 1600.
  4. Table 7, the title is incorrectly formatted. I suggest the author center the title like table 8.
  5. Table 10, the number of copies overlaps with the number of rows, so it is suggested to modify it.
  6. Line479: it is recommended to explain the meaning of Dy, if the initial letter should be capitalized.
  7. The manuscript uses tomato as a case of whether there are other plants that can be used as examples of the advantages of bio-fertilizers over mineral fertilizers, and whether the authors have conducted experiments on other species of plants to verify the results of this manuscript.
  8. The title of the manuscript does not elaborate on which part of the biofertilizer is superior to the mineral fertilizer, and it is suggested to refine the topic.
  9. The manuscript does not elaborate on whether bio-fertilizer can improve the quality of tomatoes, etc., and suggests that several physiological indicators be selected for measurement, with a view to more fully explaining the advantages of bio-fertilizer.
  10. Figure1 and figure2 do not mark significance, and it is recommended to mark significance.
  11. The title of the ordinate of figure1 is not normative,I suggest that the author standardize the ordinate title.
  12. The 31st and 33rd references in the references are different from other reference formats, and it is recommended that the format be modified after a thorough examination of the references.
  13. In the second part, two 2.1s appear in Materials and Methods, and it is recommended to modify the second 2.1 to 2.2.
  14. There are some grammatical errors.

Reviewer 2 Report

Bio-fertilizers based on digestate and biomass ash as an alternative to commercial fertilizers- the case of tomato” is an interesting study and suitable for publication after minor revisions.  I have just a few comments and suggestions indicated below:

  • Line 108: authors referred to “biomass ash and digestate” but they didn’t describe which is the biomass… is it from crop plants, cellulosic biomass or what else?
  • Lines 114-116: add a space before the brackets. Check throughout the manuscript.
  • Line 141: “S0 “ is for sulfur? and why phosphoric rock was added in fertilizers C?
  • Line 183: how long plants were dried at 65 °C?
  • Lines 216-218: Data refers to table A2, but it's not indicated in the text.
  • Lines 286- 288: In the text, authors stated that “For the plots fertilized with the b fertilizer, it was the highest on the plot with 800 kg ha-1 and for  400 kg ha-1 for the C plots”. Looking at table 6, however, the highest TFN for fertilizer B is 1600 kg ha-1. Please, check it.
  • Lines 306-309: “On average, the EX content for A, B and C fertilizers, was at the level of 4.44, 4.51 and 4.55 g kg-1, respectively. The EX content on the AC and MC was on the same level, amounting to 44.9 g kg-1. The highest EX content was obtained 308 on plots fertilized with the C fertilizer at a rate of 800 kg ha-1 (46.4 g kg-1)”.

The authors stated that EX content for A, B, C fertilizers is similar to the controls, but in the controls the EX content is ten times higher; data from the text differ from table 7. Please, check it.

-Lines 470-479: I suppose there is a mistake in text/table formatting: line numbers overlapped the table and the sentence starting from line 479 missed something.

Round 2

Reviewer 1 Report

Taking tomato as an example, this manuscript explored the feasibility of replacing commercial fertilizer with bio-fertilizer based on digestate and biomass ash, and studied the effects of bio-fertilizer on tomato yield, yield components, contents of organic compounds and minerals in fruits compared with standard inorganic fertilizer. However, the author of the manuscript did not modify it completely according to the comments of the reviewer. The following problems still exist:

1, The question of “The introduction part lacks the description of the research background in this field, and more emphasis is on the repeated elaboration of the concept and significance of biological fertilizer” was raised in the initial review, but it was not replied by the author, and the “Introduction”was not modified in the manuscript.

2, The footnote to Table 1 does not indicate the meaning of VH.

3, According to the reviewer's tenth recommendation: Figure1 and figure2 do not mark significance, and it is recommended to mark significance. The authors only modified Figure 1 and did not complete the significance analysis for Figure 2.

4, The line spacing of references 37, 38 and 39 is incorrect, please modify.

5, There is no trace of grammatical modification to the full text.
